# The Prevalence of *Chlamydia trachomatis* and Three Other Non-Viral Sexually Transmitted Infections among Pregnant Women in Pemba Island Tanzania

**DOI:** 10.3390/pathogens9080625

**Published:** 2020-07-31

**Authors:** Naomi C.A. Juliana, Saikat Deb, Sander Ouburg, Aishwarya Chauhan, Jolein Pleijster, Said M. Ali, Servaas A. Morré, Sunil Sazawal, Elena Ambrosino

**Affiliations:** 1Institute for Public Health Genomics (IPHG), Department of Genetics and Cell Biology, Research School GROW (School for Oncology & Developmental Biology), Faculty of Health, Medicine & Life Sciences, University of Maastricht, 6229 ER Maastricht, The Netherlands; n.juliana@maastrichtuniversity.nl (N.C.A.J.); samorretravel@yahoo.co.uk (S.A.M.); 2Public Health Laboratory-Ivo de Carneri, Chake Chake, Pemba Island, Tanzania; saikatdeb@gmail.com (S.D.); said@phlidc.org (S.M.A.); 3Centre for Public Health Kinetics, New Delhi 110024, India; chauhan.aishwarya2@gmail.com (A.C.); ssazawal@jhu.edu (S.S.); 4Laboratory of Immunogenetics; Department of Medical Microbiology and Infection Control, Amsterdam UMC, Vrije Universiteit, 1105 AZ Amsterdam, The Netherlands; s.ouburg@amsterdamumc.nl (S.O.); j.pleijster@amsterdamumc.nl (J.P.)

**Keywords:** *Chlamydia trachomatis*, *Neisseria gonorrhoeae*, *Trichomonas vaginalis*, *Mycoplasma genitalium*, sexual and reproductive health, pregnancy, Tanzania, sub-Saharan Africa

## Abstract

Efforts to map the burden of infections globally have shown a high prevalence of genital infections, including *Chlamydia trachomatis*, in sub-Saharan Africa. This retrospective study aimed to investigate the prevalence of selected non-viral genital infections among pregnant women in Pemba Island, Tanzania. Vaginal swabs were collected during pregnancy and stored in eNAT buffer. Detection of *C. trachomatis*, *Neisseria gonorrheae*, *Trichomonas vaginalis*, and *Mycoplasma genitalium* pathogens was performed by PCR using validated detection kits. Vaginal samples of 439 pregnant women between 16 and 48 years were tested. In fifty-five (12.5%) of them, at least one genital pathogen was detected. The most prevalent pathogen was *T. vaginalis* (7.1%), followed by *C. trachomatis* (4.6%) and *M. genitalium* (2.1%). None of the vaginal samples tested positive for *N. gonorrheae*. Consequently, among positive samples, 7.3% were for *C. trachomatis* and at least one other genital pathogen. This study provides insights on the burden of the four studied genital infections, and on the coinfections among pregnant women in Pemba Island, Tanzania. These results offer a starting point that can be useful to design further research in the field of maternal and child health in Pemba Island.

## 1. Introduction

Sexually transmitted infections (STIs) are one of the most prevalent communicable diseases globally [1]. Every day, more than 1 million such infections are reported worldwide, with varying prevalence per region [2]. Between 2009 and 2016, the sub-Saharan African region bore 40% of the global burden of STIs, with a prevalence of 20.2% curable STIs (*Chlamydia trachomatis*, *Neisseria gonorrhoeae*, *Treponema pallidum*, and *Trichomonas vaginalis*) among women of reproductive age [1,3]. About 5.1 million (2.6%) women in sub-Saharan Africa were infected with the most common bacterial STI, *C. trachomatis*, in 2008 [4]. This Gram-negative intracellular bacterium can affect (uro)genital epithelial cells, can escape the host immunological system, and cause pathology to the reproductive system, possibly resulting in cervicitis, pelvic inflammatory disease, and tubal factor infertility. During pregnancy, the infection is a risk factor for complications and adverse pregnancy outcomes, including preterm delivery, low birth weight, and postpartum sepsis [5,6,7]. The consequences of *C. trachomatis* infection on pregnant women are higher in women at risk of, or having, complicated pregnancies [6]. Therefore, *C. trachomatis* infection is particularly burdensome on residents of low- and middle-income countries, were the burden of adverse pregnancy outcomes is already high [2]. However, in approximately 60 to 80% of the cases, the infection with *C. trachomatis* might not manifest and remains asymptomatic [8]. This is unfortunate since the treatment strategy in most low-income countries has a syndromic approach, making it challenging to diagnose and eradicate this pathogen [8]. Once *C. trachomatis* or another STI are present, the host is more susceptible to the acquisition and transmission of other bacterial, protozoan, and viral STIs, including by the human immunodeficiency virus (HIV) [9].

Other treatable and mostly asymptomatic STIs, such as by *Neisseria gonorrhoeae*, *Trichomonas vaginalis*, and *Mycoplasma genitalium*, can also cause serious pathology of the reproductive system if left untreated, and have been independently associated with various adverse pregnancy outcomes [10,11,12,13,14,15]. Similarly to *C. trachomatis*, these pathogens are highly prevalent in sub-Saharan African countries, where limited research has evaluated the association between maternal infection and adverse pregnancy outcomes in these populations [4]. In addition, older studies might have utilized culture-based methodologies, whereas more recent ones employ the more sensitive molecular testing techniques to test for pathogens presence [10,11,14].

In 2016, the World Health Organization (WHO) developed the Global Health Sector Strategy on STI 2016–2021 that provided vision, goals, targets, and strategies for the prevention, control, and management of sexually transmitted infections [3]. In order to reach the Strategy’s goal of “ending STI epidemics as major public health concern”, five directions were suggested [3]. In line with the first direction, “information for focused action”, more data on STI burden, including STI prevalence estimates, are needed for both general and high-risk populations in urban and rural settings [3,16].

In their review, Adachi et al. suggested that the individual studies’ prevalence rates between 0 and 31.1% for *C. trachomatis* infection in pregnant women from sub-Saharan Africa are similar, or higher, than the prevalence (2.6%) in non-pregnant women aged 15–49 years old in sub-Saharan Africa reported by the WHO [6]. Currently, only a few studies reported the prevalence of STIs during pregnancy in individual countries (Sudan, Cameroon, Democratic Republic of Congo (DRC), Gabon, Nigeria, Kenya, Uganda, Tanzania, Malawi, Zambia, Botswana, Mozambique, and South Africa) in the sub-Saharan African region [6]. A recent systematic review of antenatal clinic attenders in East Africa, including Tanzania, reported a mean prevalence of 6.8% for *T. vaginalis*, 4.2% for *C. trachomatis*, and 2.3% for *N. gonorrhoeae* infections [17]. A prevalence of 3.2% for *M. genitalium* infection has been reported in a non-pregnant Tanzanian cohort [18].

Limited data about STIs prevalence exist for the islands belonging to the Zanzibar archipelago in Tanzania, particularly Pemba Island. Pemba Island is mainly rural, with minimal access to doctors and laboratory facilities in the primary health care system [19]. To promote maternal and child health and follow the WHO Global Health Sector Strategy on Sexually Transmitted Infections 2016–2021, each country needs to define specific populations that are affected by STI epidemics based on epidemiological and social context [3]. It is important to investigate the burden of curable STIs in rural African populations, especially in pregnant women where the risk to develop health complications for both the parturient and her fetus is high. This study aimed to provide initial insights in the field by assessing the presence of *C. trachomatis*, *N. gonorrhoeae*, *T. vaginalis*, and *M. genitalium* pathogens and reporting their prevalence among a subset of pregnant women from Pemba Island.

## 2. Methods

### 2.1. Sample Collection and Design

Samples collection was performed in the context of a previously established (AMANHI) biobanking effort with the support of the Bill and Melinda Gates Foundation, that was initiated in 2014 [20]. This biobanking effort includes, among other biological samples, vaginal samples and data from women with ultrasound-confirmed pregnancy and who are more than 8 weeks of gestation at the time of sampling [20]. The vaginal swabs analyzed in this retrospective cohort study were collected between March 2018 and January 2019 under the supervision of a health worker in health care facilities in Pemba Island. The vaginal swabs were stored in 1 mL eNAT buffer (Copan Italia, Brescia, Italy) at −20 °C at the Public Health Laboratory—Ivo de Carneri in Pemba Island. The participants filled a questionnaire before sample collection, on baseline sociodemographic data and health information about the current and previous pregnancies. For the purpose of this study, the vaginal sample collected at the earliest timepoint from each participant was selected to be tested for this study. The mean gestational age (GA) at sampling was 15.5 ± 6 weeks. The GA range of collection was 8–40 GA weeks. Samples were transported in dry-ice to the Netherlands and stored at −20 °C until further processing. In the context of the previously established biobanking effort, all the participants gave their informed consent to samples and data collection, and an ethical approval for the samples’ use was obtained from the Zanzibar Medical Research and Ethics Committee (ZAMREC) [20].

### 2.2. DNA Isolation and Real-Time PCR

DNA from 439 vaginal swabs was extracted with the Chemagen (Perkin-Elmer, Baesweiler, Germany) automated DNA extraction machine by using the buccal swab extraction kit according to the manufacturer’s instructions [21]. In short, 66.7 μL of eNAT swab suspension per sample was defrosted and 133.3 μL Chemagen lysis buffer was added. The vaginal swab eNAT/Lysis mix was vortexed with 200 μL Chemagen lysis buffer and 10 μL Proteinase K. In addition, 5 μL of internal amplification control (IAC) of the CE-IVD-certified Presto test (Goffin Molecular Diagnostics, Houten, The Netherlands) was added to the eNAT/Lysis mix. The mix was incubated at 56°C while being shaken at 450 rpm for 15 min before being placed in the Chemagen machine for the washing and elution process [21]. Samples were later stored at 5 °C.

Afterwards, *C. trachomatis*, *N. gonorrhoeae*, and *T. vaginalis* were detected by their respective CE-IVD-certified Presto *C. trachomatis*, *N. gonorrhoeae* (Goffin Molecular Diagnostics, Houten, The Netherlands), and Presto *T. vaginalis* (Goffin Molecular Diagnostics, Houten, The Netherlands) tests and real-time polymerase chain reaction (PCR) with ABI Taqman 7500 (Applied Biosystems, Foster City, CA, USA) according to the manufacturer’s instructions [22,23]. For *M. genitalium* detection, an *M. genitalium* assay, as described in Muller et al., was used on the LightCycler 480 II PCR machine (Roche Diagnostics, Basel, Switzerland) [24]. The following primers were used to target the *mg219* gene of *M. genitalium*: MG–041 (forward primer) 5′-CGG ATC AAG ACC AAG ATA CTT AAC TTT–3′ and MG–042 (reverse primer) 5′- AGC TTG GGT TGA GTC AAT GAT AAA C- 3′ together with probe MG–48 (5′–6 FAM-CCA GGG TTT GAA AAA GCA CAA CAA GCT-BHQ1–3′) (Biolegio BV) [24]. The mastermix of the in-house *M. genitalium* assay consisted of 12.5 μL of SensiMix (Bioline, Australia), 0.75 μL of 300 nM Forward and 0.75 μL of 300 nM Reverse Primer, 0.5 μL of 200 nM Probe, and 0.5 μL of PCR water. The LightCycler 480 II instrument was programmed as follows: after an initial denaturation of 10 min at 95 °C, 45 cycles were performed consisting of 15 s at 95 °C and 1 min at 60 °C. After each cycle, a single fluorescence reading (FAM^TM^ at 465–510 nm) was taken. Color compensation objects were created as described in the LightCycler^®^ 480 Operator’s Manual.

A total of 15 μL of the mastermix was added to 10 μL DNA extraction for the PCR detection.

The positive controls of *C. trachomatis*, *N. gonorrhoeae*, *T. vaginalis*, and *M. genitalium* pathogens consisted of different concentrations of their DNA that were isolated from standardized cultures for these microorganisms. Water was used as negative control in each PCR run. Samples were tested in duplicates for *C. trachomatis*, *N. gonorrhoeae*, and *T. vaginalis*. They were considered positive if the crossing point value (Cp) was between 11 and 38. In addition, samples with a Cp value of 39–40, but an S-shape amplification curve, were also considered positive. No replicates were performed for *M. genitalium* analysis, but the same Cp value thresholds were considered for determination of the positivity. If the duplicates were discordant or the amplification curve was not S-shaped (in case of *M. genitalium*, only the latter applied), a third (or second for *M.genitalium*) PCR reaction was performed to determine the definite result.

### 2.3. Statistical Analysis

The sample size for this study was calculated beforehand using the formula by Daniel WW [25], with a population size of 19,744 pregnant women as recorded in 2018 in Pemba [26], and precision was set at 0.05 and a 95% level of confidence interval (CI). The adequate sample size estimation was 62, 97, 35, and 48 pregnant women for *C. trachomatis*, *N. gonorrhoeae*, *T. vaginalis*, and *M. genitalium*, respectively. Thus, 242 pregnant women were needed to estimate the combined prevalence of these curable genital infections in Pemba Island, Tanzania. The 95% CI for each proportion was also calculated.

To compare dichotomous data, Fisher’s exact test was performed with IBM SPSS Statistics version 25 (IBM, Armonk, NY, USA). Odds ratios (OR) with 95% confidence intervals (CI) are provided for the association between maternal age or school attainment and infection status. A *p*-value of less than 0.05 was considered statistically significant.

## 3. Results

### 3.1. Characteristics of the Study Population

All 439 tested women were of Shirazi ethnic origin, and 99.8% of them were Muslims. The mean and median maternal age was 28 years (range 16–48) (Table 1). Most of the women were multigravida (84%), and the minority reported to have had an eventful obstetric history with stillbirth (8.0%), pre-rupture of membranes (PROM) (4.9%), or preterm delivery (7.0%). One woman reported that she had been previously diagnosed with human immunodeficiency virus/acquired immune deficiency syndrome (HIV/AIDS).

### 3.2. Genital Infections Prevalence

In total, 55 of the 439 (12.5%) vaginal samples tested positive for one or more genital pathogens (Figure 1). The prevalence of *C. trachomatis* infection was 4.6% (95% CI 2.8–6.9%), of *T. vaginalis* 7.1% (95% CI 4.8–9.9%), and of *M. genitalium* 2.1% (95% CI 0.9–3.9%) in vaginal samples of this pregnant cohort. *N. gonorrhoeae* was not detected in any of these vaginal samples. In four vaginal samples, two or more STI pathogens were detected (Figure 1). Among them, two vaginal samples resulted positive for *C. trachomatis* and *T. vaginalis* coinfection, one vaginal sample for *C. trachomatis* and *M. genitalium* coinfection, and one vaginal sample for *C. trachomatis*, *T. vaginalis*, and *M. genitalium* coinfection. One of the women whose vaginal sample tested positive for *C. trachomatis* and *T. vaginalis* coinfection also self-reported in the questionnaire that she had been previously diagnosed with HIV/AIDS.

Pathogens were mostly detected in vaginal samples of women between 25–34 years old (Figure 2), which was also the largest group. The risk of infections associated to different age groups (15–24 years, 25–34 years, and 35–49 years) was tested. Overall, no significant difference was observed (data not shown), except for a three times higher risk of *T. vaginalis* infection in women in the older age group (35–49 years), compared to women between 15 and 24 years (OR = 0.29, 95% CI 0.020–0.88, *p* value = 0.03).

Five women did not report the number of years they attended school. Overall, results from this study show that the prevalence of detected infections in samples from women who attended school for more than three years was lower compared to women who attended school up to two years, which was also the largest group. Such a difference was not significant (data not shown).

Twelve of the 371 women who filled in the question of urinary tract infections (UTI) symptoms in the questionnaire reported that during the ongoing pregnancy, they had been having symptoms of UTI. Of these women, one of the vaginal samples tested positive for *C. trachomatis* infection and another vaginal sample for *T. vaginalis* infection. However, no association was found between these symptoms and having a genital pathogen (*C. trachomatis*: *p*-value = 0.44; *T. vaginalis*: *p*-value = 1.0). In total, 404 women whose samples were used in this study filled the antibiotic use question in the questionnaire. Thirteen women self-reported antibiotics use during their ongoing pregnancy. None of their vaginal samples tested positive for either one of the four genital pathogens.

## 4. Discussion

To our knowledge, this is the first study that reports the burden of these selected STIs in samples from a group of pregnant women living in Pemba Island, Tanzania. In the tested vaginal samples, the prevalence of *T. vaginalis* infection (7.1%) was the highest, followed by *C. trachomatis* (4.6%) and *M. genitalium* (2.1%) infections.

Both *T. vaginalis* and *C. trachomatis* infections’ prevalence is within the same range of most reported studies from the East African region, which is respectively 6.8% (95% CI 4.6–9.0) for *T. vaginalis* and 4.2% (95% CI, 2.8–5.6) for *C. trachomatis* infection [17]. The burden of *T. vaginalis* and *C. trachomatis* genital infections in pregnant women living in Pemba seems to be also in concordance with the burden found among antenatal clinic attendees in two urban cities in mainland Tanzania, namely Tanga and Dar es Salaam [27,28]. Chidou et al. retrospectively analyzed the STI prevalence based on HIV status in pregnant women between 18 and 44 years old attending antenatal care clinics in Tanga city [28]. HIV-positive pregnant women, compared to HIV-negative ones, had a lower prevalence of *C. trachomatis* (0% vs. 3%) and higher prevalence of *T. vaginalis* (18.8% vs. 5%) infection [28]. The current study did not assess the HIV status in the tested samples. Nonetheless, older governmental surveillance data from 2000 showed a low HIV infection prevalence (0.93%) in pregnant women from Pemba [29]. In line with this, the data from the current study relate the most to Chidou et al.’s results from the HIV-negative group. Between 2001 and 2003 in Dar es Salaam, one of the largest urban cities of East Africa, the prevalence of *C. trachomatis* infection was 3.5% and *T. vaginalis* infection was 4.2% in a cohort of pregnant HIV-positive women that were mostly (>68%) between 20 and 30 years old [27]. However, Chidou et al. and Aboud et al. argued in their paper that the unexpected low trichomoniasis prevalence might be due to the laboratory-based diagnostic approach and use of a low insensitivity microscopic examination, instead of the syndromic approach and molecular-based diagnostic techniques [27]. This present study used Presto PCR-based test kits, which have been shown to have high sensitivity and specificity for the detection of *C. trachomatis*, *N. gonorrhoeae*, and *T. vaginalis* [22,23]. *N. gonorrhoeae* was not detected in this studied cohort. Similar to the prevalence of *N. gonorrhoeae* infection in this study, the prevalence was also low in the other pregnant cohorts in Dar es Salam (0.2%) and Tanga (0% in the HIV-negative cohort and 3.5% in the HIV-positive group) [27,28].

In contrast with the results from this study, the prevalence for *T. vaginalis* (12.4%), *C. trachomatis* (11.4%), and *N. gonorrhoeae* (6.7%) infections is higher among a cohort of 403 pregnant adolescents between 15 and 19 years old living in rural areas of Mwanza in mainland Tanzania [30]. Both study settings are rural sites in Tanzania, but the number of tested samples, age difference, and religious practices might help explain the difference with the results presented here. Firstly, in this current study, the median age (28 years) was above the adolescent age range and only 33 women were tested in the age range category of 15 to 19 years old. Among them, two vaginal samples were positive for *T. vaginalis* and one for *M. genitalium* (data not shown in tables). Previous studies from the United States of America, the Netherlands, China, and other sub-Saharan African countries also have shown that adolescents (<25 years) have a higher risk of acquiring STIs due to their biological susceptibility to STIs, sociological, psychological, and behavioral factors [30,31,32,33,34,35,36]. However, religion and cultural differences might also play a role, Mwanza city has a predominantly Christian rural population, while Pemba Island has a predominately rural community of Afro-Shirazi Muslims. Sociologists argue that the predominance of Muslim religion may have an impact on sexual behavior, particularly less pre- and extramarital sex, that are risk-factors for acquiring STIs [37,38]. This study also observed that pregnant women between 35 and 49 years old in Pemba have a higher risk of having an *T. vaginalis* infection than women between 15 and 24 years old. This might be explained because of the natural history of some pathogens and the ability of the human body to clear *C. trachomatis*, *T. vaginalis*, *M. genitalium*, and *N*. *gonorrhoeae*. Due to the treatment options and ethical considerations, data about the natural course of these curable STI pathogens, especially during pregnancy, are scarce. However, earlier studies have shown that non-pregnant women can have an average duration of *T. vaginalis* for at least three–five years, while other studies show a faster resolution for *C. trachomatis* (45% of non-pregnant women cleared in one year and 44% pregnant women cleared in 1–14 weeks)*, M. genitalium* (median clearance was 2.1 months), and *N. gonorrhoae* (25% of asymptomatic women cleared over 5.4 months) [39,40,41,42,43]. Therefore, it is hypothesized that the slower host defense clearance for *T. vaginalis* might explain how the point prevalence for *T. vaginalis* infection of women with a longer reproductive history (older women) might be higher than women with a shorter one (younger women), particularly when individuals were not treated. Even though there is good treatment for these four mostly asymptomatic STIs, further (retrospective) research on the clearance of these pathogens is needed to determine the long-term effect of pathogens on sexual, maternal, and fetal health. This information might be particularly important for rural populations, where medication acquirement is more difficult.

Unfortunately, data from *M. genitalium* infections among pregnant women in Tanzania, or even East Africa, are scarce. The prevalence of 2.1% (95% CI 0.9–3.9) for *M. genitalium* found in this study corresponds with the prevalence found in a non-pregnant Tanzanian female cohort between 20 and 44 years old (3.2%; 95% CI 2.8–4.2) [18]. Similar prevalence for *M. genitalium* infection was observed in a pregnant cohort of women between 18 and 45 years old in Kenya, that reported a prevalence of 6.3% (95 CI% 2.1–14.2 %) [44]. Both studies in Tanzania and Kenya also used molecular diagnosis to detect the presence of *M. genitalium*, namely real-time and quantitative PCR, respectively.

The present study identified genital coinfections in 7.3% of the tested-positive vaginal samples among pregnant women in Pemba Island. Interestingly, all of the vaginal samples with multiple pathogens infection were positive for the *C. trachomatis* one. Previous studies mostly researched the interaction between *C. trachomatis* and *N. gonorrhoeae* solely and did not consider including other sexually transmitted pathogens in their analysis [45,46]. From these previous studies, the exact pathogenesis of *C. trachomatis* in the pathogen–pathogen and pathogen–host interaction is still not well understood [45]. However, this study suggests that coinfections of *C. trachomatis* with other pathogens, such as *M. genitalium* or *T. vaginalis*, might be more prevalent than the *C. trachomatis* and *N. gonorrhoeae* coinfections in some cohorts. This should also be considered when analyzing pathogen–pathogen interaction, host–pathogen response, and both maternal as neonatal health outcomes in these populations [45,47].

There are multiple strengths of this study, for instance, that the vaginal samples were collected in a biobank setting, where large-scale biospecimens have been collected in a harmonized way [20]. Furthermore, unlike other epidemiological studies that overestimate the true prevalence because their cohort includes data from patients seeking medical help due to symptoms, this study included samples collected from pregnant woman enrolled within their communities across the island, irrespective of their symptoms or medical history. In previous studies, mostly the prevalence of *C. trachomatis* and *N. gonorrhoeae* during pregnancy is reported, whereas in this study, two other pathogens, *M. genitalium* and the most prevalent non-viral STI globally *T. vaginalis*, were also investigated [40]. Additionally, the use of molecular diagnostics techniques for the detection of these four pathogens, regardless of the symptoms is also a strength of this study, since these techniques are more sensitive than Gram staining cultures or wet mount microscopic methods used by earlier studies, and are more specific than syndromic testing [48].

However, this study has also some limitations. Firstly, the questionnaire filled by participating women did not specifically ask for STI symptoms, but solely for UTI ones. Often women find it difficult to differentiate between STI and UTI symptoms, especially during pregnancy [49]. In this cohort, only a low number of women (3.2%) reported that they experienced UTI symptoms during the current pregnancy, indicating that urogenital tract symptoms were not highly prevalent. Furthermore, a low number of women (3.2%) used antibiotics during their pregnancies, indicating that the majority of women did not seek or receive any medical help for urogenital symptoms during their current pregnancy. Moreover, the use of antibiotics during the current pregnancy was not an exclusion criterion in the selection of samples for this study, and not all women filled this question in their questionnaire. Thus, the prevalence data of the tested STIs, that are cleared under antibiotic treatment, might have been underestimated.

Education has been considered a protective factor against engagement in high-risk sexual behavior and STI diagnosis [50,51]. Data from the questionnaire show that there is a small range (one–five) in schoolyears attendance between participants and more than 80% of women reported that they attended school for one or two years only. In the group attending up to two years of school, there were more samples positive for three tested genital infections compared to the women that went to school between three and five years. Nonetheless, there was no statistical association between a shorter school attendance and *C. trachomatis*, *N. gonorrhoeae*, or *M. genitalium* infection. Due to the smaller number of women in the three–five years attendance group, it remains interesting to further analyze the importance of attending school more than three years in relation to sexual and reproductive health. Additional socio-economic factors associated with STIs in other studies, including marital status and household income, are not investigated in this study [52,53]. Other sociological baseline characteristics (ethnicity, religion, smoking status) reported in this study, and additional characteristics of Pemba’s population by the Tanzania Demographic and Health Survey and Malaria Indicator Survey in 2015–2016 (low use of health insurance, low teenage motherhood, low use of contraceptives) also suggest that the population studied here is quite homogeneous. Socio-economic factors and marital status might follow the same homogenous trend [54].

In 2018, the reported stillbirth rate in Pemba Island was 25.7 (23.5–27.9), and the neonatal mortality rate was 16.0 (14.3–17.8) per 1000 births [26]. As previously mentioned, these four studied non-viral STIs continue to have an impact on maternal health and pregnancy outcomes in various countries [55,56]. The approach to diagnosis and managing them should align with global STI eradication strategies, depending on the available resources and epidemiological information per country [56,57]. In this low-resource setting, where antenatal and health care is limited, and maternal and infant morbidity is high, it is important to understand the burden of (curable) pathogens that might cause long-term impact on female reproductive health beyond pregnancy, causing, for instance, ectopic pregnancy or successive infertility [57,58]. Establishing the burden of these pathogens through epidemiological biobank data-driven research in (a)symptomatic populations is an important first step for the management and control of the burden of pathogen-related diseases in communities, including in Pemba Island [57]. However, more opportunities to strengthen the diagnostic capacity and further investigate the burden of pathogens, for instance, through hospitals and health care institution databases, are needed, with particular attention to the data from pregnant women between 35 and 49 years. With further research efforts, not only better medical treatment can be provided for infected women, but culture-specific educational, prevention, and awareness strategies can be adjusted or be created with the relevant Ministry of Health. Evidence of STIs among pregnant women in this island warrants a follow-up on the effects of such genital pathogens on adverse pregnancy outcomes in this population.

In conclusion, this study detected an overall 12.5% prevalence for three out of the four studied non-viral STIs in vaginal samples from a homogeneous pregnant population in Pemba Island, Tanzania. In order to improve health outcomes in mothers and children, and align with the WHO’s Global Health Sector Strategy on STI goals, it is essential to not solely monitor the epidemiology of *C. trachomatis*, *T. vaginalis*, *N. gonorrhoeae*, and *M. genitalium* infections at a local and regional level, but also detect coinfections and understand their role in adverse pregnancy outcomes. The present study provides the first evidence and awareness of the burden of genital pathogens within Pemba Island, Tanzania.

## Figures and Tables

**Figure 1 pathogens-09-00625-f001:**
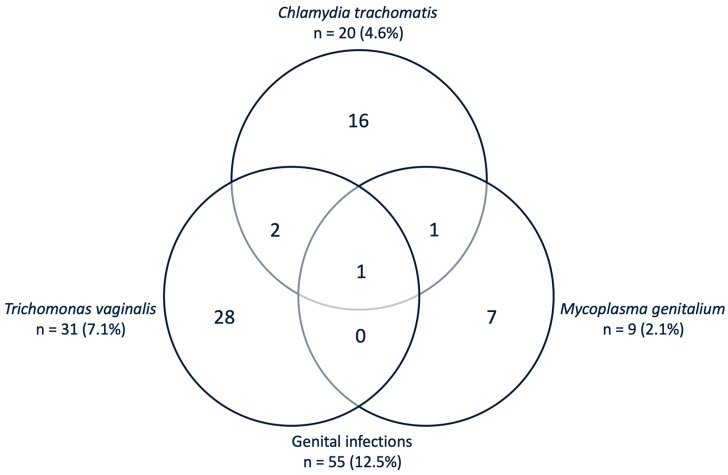
Venn diagram with absolute numbers and prevalence (%) of infections in tested vaginal samples from 439 pregnant women.

**Figure 2 pathogens-09-00625-f002:**
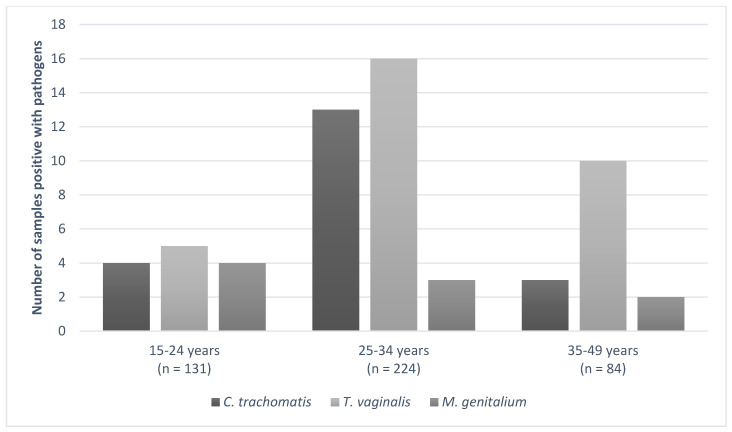
Distribution of genital pathogens among maternal age groups.

**Table 1 pathogens-09-00625-t001:** Baseline characteristics of the study participants.

	No (%) or Mean (Range)
**Mean maternal age (n = 439)**	28.3 (16–48) year
**Mean gravidity (n = 434)**	4.6 (1–16)
**Mean parity (n = 374)**	3.6 (0–10)
**Number of first pregnancy (n = 374)**	60 (16.0%)
**History with stillbirth (n = 374)**	30 (8.0%)
**History with PROM (n = 371)**	18 (4.9%)
**History with preterm delivery (n = 371)**	26 (7.0%)
**Previously diagnosed with HIV/AIDS * (n= 430)**	1 (0.23%)
**Number of years attended in school (n = 434)**	2.1 ± 1.7 (1–5) (Mean ± SD (range)) n = 127 (29.3 %) one year n = 241 (55.5 %) two years n = 13 (3.0 %) three years n = 3 (0.7 %) four years n = 50 (11.5%) five years
**Smoking (n = 431)**	0 (0%)
**Ethnicity (n = 439)**	100% Shirazi (Zanzibar Africans)
**Religion**	
- Islam	433 (99.8%)
- Christian	1 (0.2%)

HIV/AIDS: human immunodeficiency virus/acquired immune deficiency syndrome. PROM: pre-rupture of membranes. * HIV/AIDS status is based on self-reporting.

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
