# Peer review of "The Prevalence of Chlamydia trachomatis and Three Other Non-Viral Sexually Transmitted Infections among Pregnant Women in Pemba Island Tanzania"

_pathogens, 2020, doi:10.3390/pathogens9080625_

Round 1

Reviewer 1 Report

Dear Authors,

The authors of this manuscript describe the prevalence of the main non-viral sexual transmitted infections (STIs) among pregnant women in Pemba Island Tanzania (N=439, median age 28 years, mostly Muslims), using validated PCR kits: Trichomonas vaginalis 7.1%, Chlamydia trachomatis 4.6%, Mycoplasma genitalium 2.1% and Neisseria gonorrhoeae none infection.

I congratulate the authors for this interesting and well permorfed study. With the aim of improving the Discussion, I encourage the authors to think about two questions: 1- To describe the prevalence of C. trachomatis infection by age ranges, in order to compare it not only with other studies from Africa, but also with others from developed countries with higher median age in pregnant women; 2- To suggest possible prevention strategies that could be implemented in undeveloped countries to minimize the burden of STIs on health outcomes in pregnant women and their children.

In this way, I attach below some minor comments.

1. Lines 90-91: Please, add the period of the study and if there was any selection of the population studied.

2. Line 139 and table 1: Due that mean and median maternal age is lower that the usually described in studies of other developed countries, it could be of interest to describe the distribution of the study population by age ranges (<25, 25-30, >30 years).

3. Line 149: In the same way and due that C. trachomatis infection prevalence has been described higher in younger pregnant women, please add the prevalence data by age groups, in order to compare the results with other studies.

4. Line 163: I think that this sentence (“Furthermore,…”) would be better understood after the next one (“In total,…”). Or simply, “Thirteen out of 404 women reported antibiotics use during pregnancy.”

5. Lines 187-189: Median age of the population in this study?

6. Lines 202-204: Ok. Because of this, and to compare prevalences between different studies, please add prevalence of infection by age ranges. In this context, it would be of interest to compare these prevalences with other studies outside Africa (perhaps a table with prevalences described in different countries by age ranges, or cite one study with a similar table with this comparations?).

7. Lines 230-232 and 267-270: With this data, would you recommend a screening of some STIs in pregnant women of this population, or any other posible prevent strategy in undeveloped countries? If there will be economic constraits and to minimize the costs, in what age ranges? In this context, I think the comment in lines 267-270 of the conclusion should be more feasible.

I hope all these suggestions will improve this manuscript.

Kind regards.

Author Response

Reviewer 1

Dear Authors,

The authors of this manuscript describe the prevalence of the main non-viral sexual transmitted infections (STIs) among pregnant women in Pemba Island Tanzania (N=439, median age 28 years, mostly Muslims), using validated PCR kits: Trichomonas vaginalis 7.1%, Chlamydia trachomatis 4.6%, Mycoplasma genitalium 2.1% and Neisseria gonorrhoeae none infection.

I congratulate the authors for this interesting and well performed study.

A: We thank the reviewer for this positive feedback! We appreciate the effort of reading our manuscript and providing positive and constructive feedback.

With the aim of improving the Discussion, I encourage the authors to think about two questions: 1- To describe the prevalence of C. trachomatis infection by age ranges, in order to compare it not only with other studies from Africa, but also with others from developed countries with higher median age in pregnant women; 2- To suggest possible prevention strategies that could be implemented in undeveloped countries to minimize the burden of STIs on health outcomes in pregnant women and their children.

A: Thank you for the insightful suggestion! We adapted the manuscript as suggested. We not only described the distribution of the study population by age ranges, but also assessed whether age is a risk factor for the tested infections. We additionally looked at educational attainment as risk factors for the tested infections.

Unlike in other populations, the odds of T. vaginalis infection prevalence was 3x higher in women of older age group (>35 years) compared to younger women (<25 years). In our discussion we elaborated on this finding and suggested a hypothesis (lines 246-263). Where possible, the age ranges were also compared with the studies mentioned in the discussion section.

Suggested further steps are also elaborated more in details in the second to last paragraph (lines 323-340).

We believe that now, because your suggestion, the manuscript provides more interesting information to the readers of Pathogens. We hope that our changes and adjustments match your expectations.

In this way, I attach below some minor comments.

  1. Lines 90-91: Please, add the period of the study and if there was any selection of the population studied.

A: this has been added in line 96.

  1. Line 139 and table 1: Due that mean and median maternal age is lower that the usually described in studies of other developed countries, it could be of interest to describe the distribution of the study population by age ranges (<25, 25-30, >30 years).

A: this has been adjusted in figure 2.

  1. Line 149: In the same way and due that C. trachomatis infection prevalence has been described higher in younger pregnant women, please add the prevalence data by age groups, in order to compare the results with other studies.

A: this has been adjusted in figure 2 and mentioned in lines 184-188.

  1. Line 163: I think that this sentence (“Furthermore,…”) would be better understood after the next one (“In total,…”). Or simply, “Thirteen out of 404 women reported antibiotics use during pregnancy.”

A: this has been revised in line 201.

  1. Lines 187-189: Median age of the population in this study?

A: Unfortunately this data is not reported from this specific study. The only available data is the number of participants in age groups (15-19 years; n=26), (20-24 years; n=142), (25-29 years; n= 153), (30-34 years; n = 77), (>35 year; n=30). In line 223 we added that the majority of the population in this study had between 20 and 30 years.

  1. Lines 202-204: Ok. Because of this, and to compare prevalences between different studies, please add prevalence of infection by age ranges. In this context, it would be of interest to compare these prevalences with other studies outside Africa (perhaps a table with prevalences described in different countries by age ranges, or cite one study with a similar table with this comparations?).

A: As you suggested, in the discussion we also elaborated our STI data based on maternal age with findings from other studies outside of Africa (lines 237-243).

  1. Lines 230-232 and 267-270: With this data, would you recommend a screening of some STIs in pregnant women of this population, or any other possible prevent strategy in undeveloped countries? If there will be economic constraints and to minimize the costs, in what age ranges? In this context, I think the comment in lines 267-270 of the conclusion should be more feasible.

A: Great point and good questions, we elaborated on this suggestion in the second to last paragraph (lines 323-340).

I hope all these suggestions will improve this manuscript.

A: They surely do, We hope to have addressed them sufficiently.

Reviewer 2 Report

The manuscript by Juliana et al. describes a survey of banked vaginal swabs from several hundred pregnant women in Tanzania. The information obtained is useful in that it provides information on prevalence of a specific set of non-viral STIs in the particular population of Pemba Island. The objective is stated and results are clearly presented. The discussion is largely a comparison of the results of this particular study with other studies of STI prevalence in African populations, which is appropriate. I have some suggestions to improve the manuscript.

Introduction

The authors suggest in both the Introduction and in the Discussion that a main motivator for the work was the WHO’s Global Health Sector Strategy on STIs. To improve the readability of the paper, the authors should elaborate on which aspects of the strategy are being specifically addressed by their study. How does this particular study of pregnant women of Pemba Island address the WHO’s priorities or goals? Adding this information will improve the presentation of context and rationale for the study.

Why is this a “pilot study”? This term is usually interpreted as meaning that a similar, but larger study is planned. A pilot is also often a way to evaluate methodology. Based on the Discussion, I don’t think that this is a “pilot study” (the authors discuss it as completed) and suggest removing this term.

Methods

The description of the methods currently lacks sufficient detail.

The authors must include a description of the controls (positive and negative) that were included in the PCR testing. Were DNA extractions controls performed? No-template controls? What was used as a positive control? These details are essential in order to assess the rigour of the testing, especially given that no positive results were obtained for one of the pathogens.

Were any replicates performed? If so, please state how many. If not, please state that no replicates were performed.

Was any internal amplification control included or was any method used to determine the quality of the DNA extracts used for screening? Without this information it is difficult if not impossible to interpret negative results (are they actually negative or was the sample of poor quality?). Please describe how DNA extracts were assessed for suitability for PCR.

Details on the Mycoplasma genitalium “in-house” assay must be provided, including the sequences of the primers and probe, the gene being targeted, and a description of how performance testing was done to verify the assay (analytical specificity and  sensitivity, repeatability). If this work was done elsewhere and is published, the authors should refer to it; otherwise it must be described in detail.

Results

Table 1 – make it clear that the previous HIV diagnosis is based only on self-report, and that subjects in the study were not tested for HIV.

Discussion

The authors do a good job of comparing their results to findings from other studies of African populations, although I think it would be good to include information on the methods used in those other studies since differences in detection might reflect technical differences as much as actual differences in occurrence of the pathogens.

As mentioned for the Introduction, there is a missed opportunity here in the concluding paragraph to more specifically describe HOW the information generated in this study contributes to the WHO’s Global Health Sector Strategy on STIs. How will it be used? What are the next steps?

Line 255 – It is unclear what “a small gap (1-5) in school years” means. Please clarify.

Additional comments

The manuscript would benefit from overall editing for improvement of English as there are numerous grammatical errors, etc.

Avoid using the term “infection” to refer to microorganisms, e.g. in the Abstract, “This pilot study provides insights on the burden of the four studied genital infections, including trachomatis…”. Chlamydia trachomatis is not an infection, it is a pathogen. This needs to be corrected throughout the manuscript.

Line 37 – “prevalence rate” not correct – delete “rate”

Line 36 – what is “African region”, why not just “Africa”?

Author Response

Reviewer 2

The manuscript by Juliana et al. describes a survey of banked vaginal swabs from several hundred pregnant women in Tanzania. The information obtained is useful in that it provides information on prevalence of a specific set of non-viral STIs in the particular population of Pemba Island. The objective is stated and results are clearly presented. The discussion is largely a comparison of the results of this particular study with other studies of STI prevalence in African populations, which is appropriate. I have some suggestions to improve the manuscript.

A: We thank the reviewer for reading our manuscript and providing positive and constructive feedback. We tried to incorporate and implement all the suggestions.

We believe that, thank to your feedback, the manuscript has now improved and will hopefully be of even more interest to the readers of Pathogens. We hope that our changes and adjustments match your expectations.

Introduction

The authors suggest in both the Introduction and in the Discussion that a main motivator for the work was the WHO’s Global Health Sector Strategy on STIs. To improve the readability of the paper, the authors should elaborate on which aspects of the strategy are being specifically addressed by their study.

A: This has been elaborated in lines 65-68.

How does this particular study of pregnant women of Pemba Island address the WHO’s priorities or goals?

A: This has been elaborated in lines 81-84.

Adding this information will improve the presentation of context and rationale for the study.

Why is this a “pilot study”? This term is usually interpreted as meaning that a similar, but larger study is planned. A pilot is also often a way to evaluate methodology. Based on the Discussion, I don’t think that this is a “pilot study” (the authors discuss it as completed) and suggest removing this term.

A: The term ‘pilot’ has now been removed

Methods

The description of the methods currently lacks sufficient detail.

The authors must include a description of the controls (positive and negative) that were included in the PCR testing. Were DNA extractions controls performed? No-template controls? What was used as a positive control? These details are essential in order to assess the rigour of the testing, especially given that no positive results were obtained for one of the pathogens.

Were any replicates performed? If so, please state how many. If not, please state that no replicates were performed.

Was any internal amplification control included or was any method used to determine the quality of the DNA extracts used for screening? Without this information it is difficult if not impossible to interpret negative results (are they actually negative or was the sample of poor quality?). Please describe how DNA extracts were assessed for suitability for PCR.

Details on the Mycoplasma genitalium “in-house” assay must be provided, including the sequences of the primers and probe, the gene being targeted, and a description of how performance testing was done to verify the assay (analytical specificity and  sensitivity, repeatability). If this work was done elsewhere and is published, the authors should refer to it; otherwise it must be described in detail.

A:  Thank you for elaborating on these points! Based on your suggestions, we wrote a more detailed description of the DNA extraction, M. genitalium and reference is provided in the method section (lines 122-129). We do not believe there was an DNA extraction issue or that that was the reason for not detecting N. gonorrhoeae (NG), since C. trachomatis was simultaneously tested in the real-time PCR and an IAC was added and controlled for in each sample (line 113). The IAC (Internal Amplification Control) is a Competitive internal control with an amplimer longer than the PCR amplicon of interest (for example NG) meaning if NG is present the Cp-value of the IAC will be undetermined or low (<31), if no NG is present the IAC will be present (Cp-value >31). If IAC was undetermined (which was never the case in these samples), it might state that there was inhibition in the PCR or the DNA isolation was not done correctly. All negative controls in each PCR run, the Cp-value of the IAC, pathogen DNA were undetermined. Information about the replicates is now also elaborated (lines 136-142). Each PCR run also consisted of wells for positive controls that were always positive (lines 134-137).

Results

Table 1 – make it clear that the previous HIV diagnosis is based only on self-report, and that subjects in the study were not tested for HIV.

A: This has been clarified in table and line 167.

Discussion

The authors do a good job of comparing their results to findings from other studies of African populations, although I think it would be good to include information on the methods used in those other studies since differences in detection might reflect technical differences as much as actual differences in occurrence of the pathogens.

A: This is indeed a good point, such information is present in lines 223-228 and 269-270.

As mentioned for the Introduction, there is a missed opportunity here in the concluding paragraph to more specifically describe HOW the information generated in this study contributes to the WHO’s Global Health Sector Strategy on STIs. How will it be used? What are the next steps?

A: This has been elaborated in lines 323-340.

Line 255 – It is unclear what “a small gap (1-5) in school years” means. Please clarify.

A: This has been clarified in lines 307.

Additional comments

The manuscript would benefit from overall editing for improvement of English as there are numerous grammatical errors, etc.

Avoid using the term “infection” to refer to microorganisms, e.g. in the Abstract, “This pilot study provides insights on the burden of the four studied genital infections, including trachomatis…”. Chlamydia trachomatis is not an infection, it is a pathogen. This needs to be corrected throughout the manuscript.

A: The grammar has been revised throughout the manuscript and the difference between pathogens and infection has been clarified.

Line 37 – “prevalence rate” not correct – delete “rate”

A: This has been removed.

Line 36 – what is “African region”, why not just “Africa”?

A: This has been revised to sub-Saharan African region.